# Bit-by-Bit: Progressive QAT with Outlier Channel Splitting for Stable Low-Bit LLMs

## Abstract

Training large language models (LLMs) at ultra–low precision remains challenging: direct low-bit quantization-aware training (QAT) often suffers from slow convergence that demands substantial training budgets, as well as quantization errors arising from heavy-tailed outlier channels and the accumulation of errors across layers. To address these issues, we present Bit-by-Bit, a progressive QAT framework with outlier channel splitting. Our approach integrates three key components: (1) block-wise progressive training that reduces precision stage by stage, ensuring stable initialization for low-bit optimization; (2) rounding-aware outlier channel splitting, which mitigates quantization error while acting as an identity transform that preserves the quantized outputs; and (3) microscaling groups with E4M3 scales to capture dynamic activation ranges aligned with OCP/NVIDIA practices. Furthermore, we exploit the nested structure of integer quantization grids to enable a single-run, once-for-any-precision model that can be directly deployed at multiple bit-widths without retraining. We conduct comprehensive evaluations under both weight-only and weight–activation quantization settings. Under W2A2 quantization, Bit-by-Bit narrows the perplexity gap with full-precision models on WikiText2 to just 2.25, consistently outperforming BitDistiller by 24.19 and EfficientQAT by 20.59 on Llama2-7b. Moreover, on the Llama3 family—known for its quantization difficulty, Bit-by-Bit surpasses other QAT baselines. Code is available in the Appendix.

## 1 Introduction

Large language models (LLMs), such as GPT-5 (OpenAI, 2025) and DeepSeek (Liu et al., 2024a), have demonstrated exceptional performance on a wide range of natural language processing tasks (Yang et al., 2019; Liu et al., 2019; Talmor et al., 2018; Chowdhery et al., 2023; Zheng et al., 2020) and have significantly improved agent capabilities in applications such as coding assistance (xAI, 2025). A key factor behind this success is the scaling law (Kaplan et al., 2020), which indicates that increasing model size consistently improves performance. However, rapid growth in parameter counts and computational requirements introduces considerable challenges: inference latency increases sharply, and high resource demands hinder efficient deployment, both in large-scale data centers and on resource-constrained edge devices. These challenges have motivated extensive research into LLM compression techniques, including pruning, low-rank decomposition, and quantization.

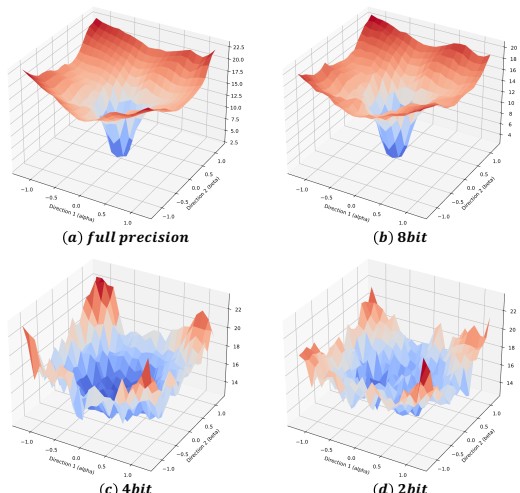

(a) *full precision*          (b) *8bit*

(c) *4bit*          (d) *2bit*

Figure 1: **Loss landscapes under different precisions.** The vertical axis denotes the loss, the horizontal axes $(\alpha, \beta)$ represent random directions in parameter space.

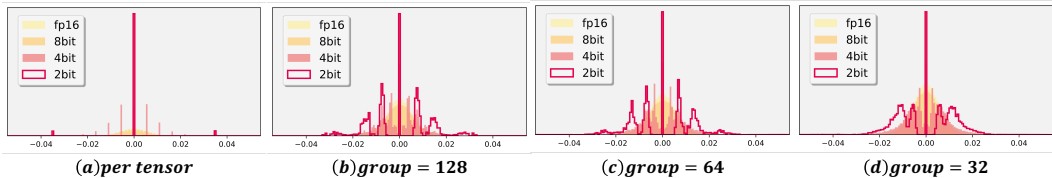

(a) per tensor    (b) group = 128    (c) group = 64    (d) group = 32

Figure 2: **Value distributions of various group granularities** showing (a) Low-bit values are nested in the high-bit grid, (b) lower bits collapse representations; larger groups improve dynamic-range.

Among various compression techniques, quantization has emerged as a particularly promising strategy. It effectively reduces model size by encoding weights and activations with fewer bits, and lowers computation by enabling low-precision arithmetic. Existing approaches fall into two families: post-training quantization (PTQ) and quantization-aware training (QAT). PTQ quantizes a pretrained model with little or no retraining and thus dominated early work; however, it often degrades sharply at ultralow precisions ($\leq$ 4-bit) (Lin et al., 2024). By contrast, quantization-aware training (QAT) incorporates the quantization process directly into the training loop to mitigate the quant error caused by low-precision representation.

To achieve low bit, existing QAT methods have explored primarily on several directions: (i) modifying the optimization objective via variants of knowledge distillation (Du et al., 2024; Chen et al., 2024a) to better align with full-precision output distributions; (ii) improving discrete gradient estimation through enhanced Straight-Through Estimators (STE) (Panferov et al., 2025; Malinovskii et al., 2024) to suppress large-error gradients; (iii) designing more robust quantizers such as clipping strategies and quantization grid (Chen et al., 2024a; Liu et al., 2025b; Du et al., 2024) to mitigate the influence of non-salient values; (iv) employing fine-grained, stage-wise schedules for learning rates and weight decay (Ma et al., 2025; 2024; Team et al., 2025); and (v) inserting orthogonal or smooth transformations (e.g., Hadamard) into training (Choi et al., 2025; Panferov et al., 2025; Tan et al., 2025; Wang et al., 2025) to reduce quantization errors introduced by outliers. Despite these advances, existing approaches still face critical stability challenges during training. They often rely on massive token budgets to converge to usable low-bit representations; demand extensive hyperparameter "wind tunnel" tuning, particularly of learning rates, since low-bit weights require larger yet inherently unstable updates; and introduce significant computational overhead from complex distillation losses, which slow training and inflate memory usage due to the need to retain both teacher and student logits. These challenges naturally raise the question: *How can we mitigate quantization error and achieve stable ultra-low-bit QAT?*

To address this, we first examine the loss landscapes under different precisions (Figure 1). We observe that as precision decreases, the loss landscape becomes increasingly uneven and discontinuous, which can trap the model in poor local minima. Moreover, this induced weight distributions are difficult to represent at low bit widths (Figure 2), making QAT optimization inherently unstable in the ultra-low-bit regime. And by further examining the quantization error across different blocks (Figure 3), we find that later layers suffer from significantly larger errors. This suggests that *the key challenge for ultra-low-bit QAT lies in the accumulation of quantization error*. So inspired by (Zhuang et al., 2018), we propose **Bit-by-Bit**, a progressive framework for stable ultra-low-bit QAT. Our main contributions are:

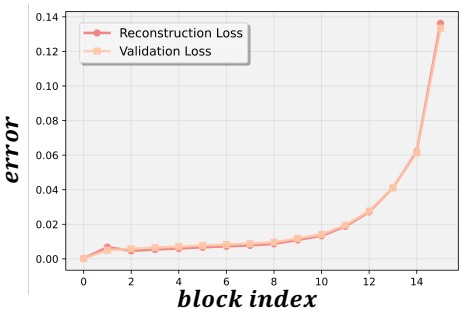

Figure 3: Layer-wise reconstruction and validation errors across Transformer blocks, showing error accumulation in later layers.

- A progressive strategy anneals precision from high to low, quantizing weights first and activations later to provide a well-conditioned start for the subsequent low-bit stage.
- Rounding-aware outlier channel splitting, which mitigates both outlier effects and rounding errors while preserving quantized outputs.
- Microscaling conventions (e.g., MXFP4, NVFP4-style), aligned with OCP/NVIDIA formats, to effectively capture dynamic ranges of full precision values.

Our comprehensive evaluation on LLaMA-2/3 and Mistral under both weight-only (w2a16) and weight–activation (w2a2) shows that BIT-BY-BIT consistently surpasses strong QAT baselines under the same training budget in ultra–low-bit regimes. On LLaMA-2 7B with w2a2 quantization, it incurs only merely +2.25 perplexity increase on WikiText2 compared to FP16 (7.72 vs. 5.47), while on LLaMA-3 family which is hard to quantize, Bit by Bit surpass other QAT methods.

## 2 RELATED WORK

### 2.1 QUANTIZATION FOR LLMS

**Post-Training Quantization (PTQ)** is a mainstream LLM compression method, with aggressive strategies down to 2-bit (Liu et al., 2024b), ternary (Kaushal et al., 2024), and binary (Gu et al., 2025). Most approaches aim to preserve a small set of salient weights to reduce error, e.g., AWQ (Lin et al., 2024) uses activation-guided scaling, SqueezeLLM (Kim et al., 2023) mixes dense/sparse formats, PB-LLM (Shang et al., 2023) combines binary and INT8, and BiLLM (Huang et al., 2024) adds residual quantization. Despite effectiveness, these designs often introduce complex implementations and kernel inefficiency.

**Quantization-Aware Training (QAT)** aims to address these issues by jointly optimizing the weights along with the quantizer to mitigate quantization error, including: LLM-QAT (Liu et al., 2023) operates without additional data but suffers from high computational overhead during teacher logits computation; QuEST (Panferov et al., 2025) filters outlier gradients and employs RMS operations combined with Gaussian and Hadamard transforms for distribution fitting; DB-LLM (Chen et al., 2024a) introduces a dual binary representation along with a deviation-aware distillation loss and BitNet (Ma et al., 2025) has demonstrated the potential of ternary weight representations, yet requires as many as 2T tokens to establish a stable low-bit model.

**Weight-Only Quantization** stores LLM weights in low precision, with recent works pushing below 1-bit representation (Gu et al., 2025; Dong et al., 2024), achieving up to $20\times$ compression. **Weight–Activation Quantization** further quantizes activations, enabling low-precision GEMM kernels and reducing IO (e.g., DeepSeek's DeepGEMM (DeepSeek, 2025)). Methods like SmoothQuant (Xiao et al., 2023a) shift quantization difficulty from activations to weights, while rotation-based approaches (QuaRot (Ashkboos et al., 2024), SpinQuant (Liu et al., 2024c)) improve robustness via orthogonal transformations. Our QAT framework supports both ultra-low-bit weight-only and weight–activation quantization.

### 2.2 GRANULARITY AND FORMAT

Quantization differs by **format**: uniform integers (fixed step), low-precision floats (non-uniform levels), and codebook-based schemes (e.g., NF4 (Dettmers et al., 2023)). It also varies by **granularity**: per-tensor, per-channel, per-group, or per-block. Recently, *micro-scaling* formats gained attention: OCP MX (MXFP4 (Rouhani et al., 2023)) shares an E8M0 scale over 32 elements, while NVIDIA NVFP4 (NVIDIA, 2025) uses 16-element blocks with E4M3 scales plus a FP32 master scale. Our method adopts this microscaling-group design to capture dynamic distributions and extends it to 2-bit quantization.

## 3 METHOD

In this section, we revisit quantization for LLMs and introduce our method, which integrates a progressive QAT strategy with Once-for-any-precision training, outlier channel splitting, and microscaling groups.

### 3.1 QUANTIZATION REVISITED

Quantization is applied to all linear layers except the LM head and the embedding layer. In group-wise quantization, the weight matrix $W \in \mathbb{R}^{m \times n}$ is partitioned into column groups of size $g$:

$$W = \left[ W^{(1)}, W^{(2)}, \ldots, W^{(G)} \right], \quad W^{(i)} \in \mathbb{R}^{m \times g}, \; G = \frac{n}{g}.$$

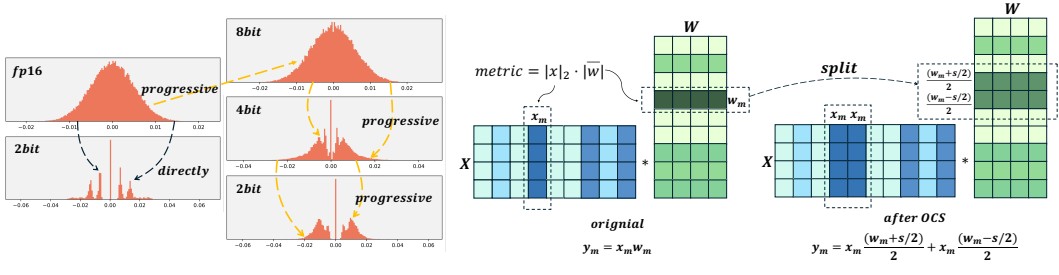

(a) *Progressive Bit by Bit QAT*    (b) *Rounding Aware outlier channel split*

Figure 4: **(a) Progressive Bit-by-Bit QAT:** Direct 2-bit QAT drives weights into coarse clusters under a non-smooth loss landscape, progressive schedule that lowers precision stage-by-stage, using the higher-precision phase to stabilize and initialize the next stage. **(b) Rounding-aware outlier channel splitting:** detect outlier channels via metric $|\mathbf{x}|_2 \cdot \overline{|w|}$, then apply identical, rounding-aware halving that keeps the quantized output unchanged.

Each group is quantized independently. For any element $x \in W^{(i)}$, we compute

$$q = \mathrm{round}\left(\frac{x}{s} + z\right),\ q \leftarrow \mathrm{clip}(q,\, 0,\, 2^n - 1),\ s = \frac{Max - Min}{2^n - 1},\ z = -\mathrm{round}\left(\frac{Min}{s}\right),$$

where $Max = \max(W^{(i)}), Min = \min(W^{(i)})$. In the following, we use the terms scale and step size interchangeably to denote $s$. Since **symmetric** quantizer can only represent three distinct levels at low bit-widths (e.g., 2-bit), or, as in strategies like SEQ (Liu et al., 2025b), map weights to a symmetric codebook such as $\{-1.5, -0.5, 0.5, 1.5\}$. We adopt an **asymmetric** quantizer with a zero-point in our methods. To incorporate the quantizer into training, we adopt the straight-through estimator (STE) to address the non-differentiability of the rounding operation. Gradients flow only through the weights, while the scale $s$ and zero-point $z$ obtained directly from closed-form expressions. No additional clipping or heuristic adjustment (Shao et al., 2023) is applied to the weights, ensuring a simple yet effective quantization scheme.

Lower-bit dequantized weights are contained in—and well-approximated by—the lattice of a slightly higher precision: for every $x_{\text{low}}$ there exists $x_{\text{high}}$ with $|x_{\text{low}} - x_{\text{high}}| \leq \frac{1}{2}s_{\text{high}}$. Hence higher-bit grids strictly refine lower-bit ones, motivating a coarse-to-fine progressive schedule.

## 3.2 PROGRESSIVELY BIT-BY-BIT QAT

As shown in Figure 1, directly optimizing at very low precision often produces a rugged loss landscape, making training susceptible to suboptimal local minima. Examining the dequantized weights further reveals that lower-bit representations collapse into only a few coarse clusters (Figure 2). Lower-bit dequantized value are contained in, and well-approximated by, the lattice of higher precision: for every $x_{\text{low}}$ there exists $x_{\text{high}}$ with $|x_{\text{low}} - x_{\text{high}}| \leq \frac{1}{2}s_{\text{high}}$. where $x_{high}$ denote the higher bit value and $s_{high}$ is the corresponding step size. This hierarchical relationship suggests a natural coarse-to-fine progression: higher-bit grids act as smooth refinements of lower-bit representations, motivating us to adopt progressive quantization as a more stable optimization path.

Directly training models at ultra-low precision is often unstable. To alleviate this issue, we adopt a progressive quantization-aware training (QAT) strategy. We begin from a relatively high precision setting, which closely matches full precision and introduces negligible quantization error, providing a well-conditioned initialization. The bitwidth is then gradually reduced across stages (e.g., from 8-bit to 4-bit and finally to 2-bit for weights), allowing the model to progressively adapt to the increasing quantization noise. For weight–activation quantization, we apply the same principle: the model is first stabilized under a configuration with low-bit weights but high-precision activations, and the activation precision is then progressively lowered in subsequent stages. This staged reduction enables the model to adapt step by step to the growing activation noise, thereby mitigating training instability. We also explored alternative progressive schedules, and further details are provided in the Appendix B.1.

Following BRECQ (Li et al., 2021) and EfficientQAT Chen et al. (2024b), we employ a block-wise objective to mitigate error accumulation. For block $i$, let $x_{w\mathbf{k}a16}^{(i)}$ denote the input activation

---

**Algorithm 1** Bit by Bit Progressive QAT and Once-for-any-precision

| | |
|---|---|
| **func** ProgressiveQAT(**Model**) | **func** Once-for-any-precision(**Model**) |

**func** ProgressiveQAT(**Model**)
1: **for** block $i = 1, \ldots, B$ **do**
2:      $y \leftarrow x^{(i)}_{w16a16} W^{(i)}_{w16a16}, \; x_{\text{ref}} \leftarrow x^{(i)}_{w16a16}$
3:      **for** $\mathbf{k} = 8$ to $2$ **do**
4:          $\mathcal{L} \leftarrow \mathbf{MSE}\left(x_{\text{ref}} W^{(i)}_{w\mathbf{k}a16}, y\right)$
5:          $x_{\text{ref}} \leftarrow x^{(i)}_{w\mathbf{k}a16}$
6:      **end for**
7:      **for** $\mathbf{k} = 8$ to $2$ **do**
8:          $\mathcal{L} \leftarrow \mathbf{MSE}\left(x_{\text{ref}} W^{(i)}_{w2a\mathbf{k}}, y\right)$
9:          $x_{\text{ref}} \leftarrow x^{(i)}_{w2a\mathbf{k}}$
10:     **end for**
11: **end for**

**func** Once-for-any-precision(**Model**)
1: **for** block $i$ **do**
2:      **for** $r \in R$ **do**
3:          $W_r = s_{share} * q^{(r)}$
4:      **end for**
5:      minimize $\mathcal{L} = \sum_{r \in R} \lambda_r \left(xW_R - y\right)$
6: **end for**
7: **schedule** $\lambda_{high} \downarrow, \lambda_{low} \uparrow$ every epoch

**func** Mapping($\mathbf{s_{share}}, \mathbf{q^h}$)
1: $q^l = q^h \gg (h - l)$
2: $W_l = s_{share} * (q^l \ll (h - l))$

---

when all preceding blocks use $\mathbf{k}$-bit weights (while activations remain FP16), and let $x^{(i)}_{w(\mathbf{k}+\Delta)a16}$ denote the activation obtained when the preceding blocks use a slightly higher precision, e.g., $w4a16$ as $w(2 + \Delta)a16$ for stabilizing $w2a16$. The full-precision reference is denoted as $w16a16$. The block-wise loss is formulated as

$$\mathcal{L}^{(i)} = \mathbf{MSE}\left[\left(x^{(i)}_{w(\mathbf{k}+\Delta)a16} W^{(i)}_{w\mathbf{k}a16}\right) - \left(x^{(i)}_{w16a16} W^{(i)}_{w16a16}\right)\right].$$

This design leverages higher-bit block activations as a more accurate teacher, improving the robustness of QAT across 8/4/2-bit regimes. A similar block-wise formulation is also applied to weight–activation quantization, where activations are progressively reduced from $a16$ to lower precisions.

**Once-for-any-precision.** Supporting multiple precisions in practice usually requires storing several models of different sizes, each obtained via separate QAT. Inspired by (Nair et al., 2025; Park et al., 2024; Cai et al., 2019), we extend our *Bit-by-Bit* framework to a unified *once-for-any-precision* paradigm, where a single model can be deployed at various bit-widths without additional retraining.

The key idea is that quantization maps a high-precision value onto a coarser grid defined by a scale factor. The most common case is mapping from floating point to integers, $w_{\text{fp}} \rightarrow s \cdot q$. However, the same principle applies between different integer precisions. Given an integer quantization at $h$ bits, $q^h$, the corresponding $l$-bit representation $2^l$ ($l < h$) can be obtained:

$$s \cdot q^{(h)} \quad \longrightarrow \quad s \cdot 2^{h-l} \cdot \left\lfloor \frac{q^{(h)}}{2^{h-l}} \right\rfloor = s \cdot 2^{h-l} \cdot q^{(l)},$$

where $q^{(l)} = \left\lfloor q^{(h)} / 2^{h-l} \right\rfloor$ is obtained by discarding the $(h - l)$ least significant bits. This shows that the $l$-bit grid is inherently nested within the $h$-bit grid. In practice, this mapping is implemented with integer bit shifts: $q^{(l)} = q^{(h)} \gg (h - l)$, $\hat{w}^{(l)} = s \cdot \left(q^{(l)} \ll (h - l)\right)$, using shift operation.

During training, we minimize a multi-precision objective $\mathcal{L} = \sum_{r \in R} \lambda_r \left(xW_R - y\right)$, where $R$ is the set of target bit-widths (e.g., $R = \{w8a16, w4a16, w2a16\}$), $\lambda_r \geq 0$ controls the contribution of each precision, $W_R$ denotes the weights quantized to $r$ bits with shared scale and $y$ denotes the full precision output. Since lower-precision grids are nested within higher-precision ones, we adopt a progressive strategy: we initially emphasize the highest bit-width to obtain a well-conditioned initialization (large $\lambda_8$), and then gradually ramp up the lower-bit losses (increasing $\lambda_4$ and $\lambda_2$) while keeping the higher-precision terms non-zero to prevent forgetting. Finally, we store the high-precision model and derive its low-precision variants via the above mapping procedure.

## 3.3 OUTILER CHANNEL SPLIT

The outlier issue has long been a major challenge in quantization, for uniform $b$-bit quantization, the step size is $s = \frac{\max(W) - \min(W)}{2^b - 1}$. Weight outliers enlarge the range $R = \max(W) - \min(W)$, thereby increasing $s$; activation outliers enlarge $\|x\|_1$. As a result, the quantization error is bounded by $\left|xW - xW_{\text{quant}}\right| \leq \frac{1}{2} s \|x\|_1$, showing that both weight and activation outliers amplify the error through range expansion and input magnitude. Prior works (Shao et al., 2023) often mitigate this problem by clipping outliers with learnable parameters. However, outliers value encode important

distributional or semantic features (Sun et al., 2024), and discarding them directly can lead to substantial performance degradation. Motivated by this, we adopt the **Outlier Channel Splitting (OCS)** (Zhao et al., 2019), which duplicates channels containing extreme activations and redistributes their contribution through an identity mapping, thereby retaining critical information while keeping the quantization process efficient.

Consider a linear layer with input vector $\mathbf{x} \in \mathbb{R}^m$, weight matrix $W \in \mathbb{R}^{m \times n}$, and output $\mathbf{y} \in \mathbb{R}^n$:

$$\mathbf{y} = \mathbf{x}W, \quad \text{where } y_j = \sum_{i=1}^{m} x_i W_{ij}.$$

Without loss of generality, assume that the last input channel $x_m$ is identified as an outlier channel. OCS duplicates the outlier channel and halves its contribution across the two copies, keeping the layer output unchanged. Formally, splitting the activation of outlier channel $m$ into two identical branches allows the output $y_j$ to be rewritten as

$$x_m W_{mj} \; \to \; \left(\tfrac{1}{2}x_m\right)W_{mj} + \left(\tfrac{1}{2}x_m\right)W_{mj} = x_m\left(\tfrac{1}{2}W_{mj}\right) + x_m\left(\tfrac{1}{2}W_{mj}\right).$$

This operation can be equivalently applied to the outlier weight row. Both formulations are mathematically identical, OCS replaces a single outlier channel with two identical copies of reduced magnitude. This operation reduces the dynamic range per channel, thereby alleviating the quantization error caused by outliers, at the cost of a small increase in channel dimensionality.

Splitting increases layer width and increase computation, so we split only a small subset of channels. For a linear layer with input $\mathbf{x} \in \mathbb{R}^m$ and weights $W \in \mathbb{R}^{m \times n}$, we define an outlier metric for each input channel $i$ as

$$metric_i = \|\mathbf{X}_i\|_2 \cdot \frac{1}{n} \sum_{j=1}^{n} |W_{ij}|,$$

where $\|\mathbf{X}_i\|_2$ denotes the $\ell_2$ norm of the $i$-th input feature aggregated across $N \times L$ tokens, and $\frac{1}{n}\sum_{j=1}^{n}|W_{ij}|$ represents the average absolute weight magnitude of channel $i$ across all output dimensions. As shown in Fig. 3, quantization error accumulates along depth, so later blocks suffer larger errors. To compensate, we adopt a *block-wise* schedule that linearly increases the split ratio with depth. Index Transformer blocks by $b = 1, \ldots, B$ from shallow to deep. For block $b$, we set

$$r_b \; = \; r_{\min} \; + \; \frac{b-1}{B-1}\left(r_{\max} - r_{\min}\right),$$

and split the top $\lceil r_b\, m \rceil$ input channels (ranked by $s_i$), where $m$ is the number of input channels in that layer. This allocates fewer splits to early blocks and more to later blocks, matching the observed depth-wise error accumulation.

For a selected outlier channel $m$ with weight row $W_{m:}$, we apply a *rounding-aware split*. Let $s$ be the (post-split) step size; we replace its contribution by two half branches with opposite half-step offsets:

$$W_{m:} \; \longrightarrow \; \left(\tfrac{W_{m:}-s/2}{2}, \; \tfrac{W_{m:}+s/2}{2}\right).$$

By nearest rounding, $Q_s\!\left(\tfrac{W_{m:}-s/2}{2}\right) + Q_s\!\left(\tfrac{W_{m:}+s/2}{2}\right) = Q_s(W_{m:})$, thus the quantized output is preserved identical. With $\mathrm{RoundErr}(z) = \left(\mathrm{Round}(z) - z\right) \in [-\tfrac{1}{2}, \tfrac{1}{2})$, the post-split error is

$$\varepsilon_{\mathrm{RA}} = x_m\left(Q_s\!\left(\tfrac{W_{m:}-s/2}{2}\right) + Q_s\!\left(\tfrac{W_{m:}+s/2}{2}\right) - W_{m:}\right) = x_m\, s\, \mathrm{RoundErr}\!\left(\tfrac{W_{m:}}{s}\right).$$

In contrast, the naive half split $(W_{m:}/2, \; W_{m:}/2)$ yields

$$\varepsilon_{\mathrm{naive}} = x_m\left(Q_s\!\left(\tfrac{W_{m:}}{2}\right) + Q_s\!\left(\tfrac{W_{m:}}{2}\right) - W_{m:}\right) = x_m\, 2s\, \mathrm{RoundErr}\!\left(\tfrac{W_{m:}}{2s}\right),$$

hence $\mathbb{E}[|\varepsilon_{\mathrm{RA}}|] = \tfrac{1}{2}\,\mathbb{E}[|\varepsilon_{\mathrm{naive}}|]$ (MSE is 1/4). If the pre-split step is $s_{\mathrm{old}}$ and splitting halves the range ($s \approx s_{\mathrm{old}}/2$), then $\mathbb{E}[|\varepsilon_{\mathrm{RA}}|] \approx \tfrac{1}{2}\,\mathbb{E}[|\varepsilon_{\mathrm{base}}|]$, while the naïve split is even with the baseline.

Table 1: Evaluation results on WikiText2 and C4 across different model sizes. Our method **Bit-by-Bit** is highlighted.

| Method | Bits | Group | WikiText2 | | | | C4 | | | |
|---|---|---|---|---|---|---|---|---|---|---|
| | | | 2-7B | 3.2-1B | 3.2-3B | 3-8B | 2-7B | 3.2-1B | 3.2-3B | 3-8B |
| FP16 | - | - | 5.47 | 9.75 | 7.81 | 6.13 | 6.97 | 12.74 | 10.44 | 8.89 |
| **Weight Only Quantization (w2a16)** | | | | | | | | | | |
| GPTQ | w2a16 | 128 | 60.5 | 2775.63 | 379.23 | 43.34 | 33.7 | 1875.41 | 323.24 | 43.28 |
| AWQ | w2a16 | 128 | 2.2e5 | 1.7e7 | 7.2e6 | 5.2e5 | 1.75e5 | 1.9e7 | 7.7e6 | 5.1e5 |
| OmniQuant | w2a16 | 128 | 11.06 | 6260.71 | 1.4e51 | 2.2e6 | 15.02 | 2442.55 | 8315.17 | 8.3e5 |
| ParetoQ | w2a16 | -1 | 10.89 | 42.82 | 26.88 | 100.04 | 12.40 | 35.08 | 24.08 | 94.97 |
| EfficientQAT | w2a16 | 128 | 7.19 | 23.89 | 14.08 | 11.31 | **8.79** | **26.09** | 18.26 | 15.26 |
| BitDistiller | w2a16 | 128 | 8.08 | 34.45 | 16.96 | 12.48 | 9.17 | 62.23 | 19.58 | 18.79 |
| **Bit-by-Bit (Ours)** | w2a16 | 32 | **6.50** | **17.07** | **11.25** | **8.87** | 9.22 | 27.40 | **17.41** | **15.18** |
| **Weight Activation Quantization (w2a2)** | | | | | | | | | | |
| SmoothQuant | w2a2 | 128 | 2.5e5 | 1.7e7 | 2.0e6 | 8.6e6 | 3.0e5 | 1.8e8 | 1.5e6 | 9.9e6 |
| SpinQuant | w2a2 | 128 | 5433.06 | 4059.73 | 4008.33 | 7931.37 | 7524.73 | 8222.23 | 8256.53 | 1.3e5 |
| ParetoQ | w2a2 | -1 | 259.74 | 1091.78 | 1018.61 | 549.71 | 135.32 | 418.22 | 401.22 | 237.21 |
| EfficientQAT | w2a2 | 128 | 26.06 | 118.24 | 56.42 | 25.86 | 23.13 | 83.85 | 51.57 | 27.27 |
| BitDistiller | w2a2 | 128 | 29.66 | 45.56 | 37.32 | 24.26 | 43.08 | 61.11 | 46.91 | 26.81 |
| **Bit-by-Bit (Ours)** | w2a2 | 32 | **7.72** | **24.99** | **14.27** | **11.54** | **15.87** | **59.75** | **26.39** | **26.45** |

## 3.4 MICROSCALING

Ultra–low-bit quantization significantly reduces computational and I/O costs, but it also severely restricts the representable dynamic range (Figure 2). To address this limitation, microscaling formats—such as MXFP4 and NVFP4, introduce a shared scale factor applied to small blocks of weights. In line with this approach, we apply per-group scaling over 32 elements and store each group scale in FP8 to minimize overhead. Unlike MX-style formats that adopt FP8 with an 8-bit exponent and no mantissa (E8M0, i.e., power-of-two scaling), our 2-bit (INT2) payload requires finer granularity than what power-of-two steps can offer. Therefore, we employ FP8 with a 4-bit exponent and 3-bit mantissa (E4M3) for group scales. This format provides sufficient mantissa precision for accurate step-size adjustment, while adding only one 8-bit scale per 32 weights, resulting in a storage overhead of just $8/32 = 0.25$ bits per weight.

## 4 EXPERIMENT

We comprehensively evaluate **Bit-by-Bit** against both post-training quantization (PTQ) and quantization-aware training (QAT) baselines. PTQ methods include GPTQ (Frantar et al., 2022), AWQ (Lin et al., 2024), OmniQuant (Shao et al., 2023), SmoothQuant (Xiao et al., 2023b), MatQuant (Nair et al., 2025), and SpinQuant (Liu et al., 2024c), while QAT baselines cover EfficientQAT (Chen et al., 2024b), ParetoQ (Liu et al., 2025b), and BitDistiller (Du et al., 2024). All experiments are run on a single H800 GPU.

### 4.1 EXPERIMENTAL SETTINGS

We test on the LLaMA (Dubey et al., 2024) and Mistral families, evaluating five zero-shot reasoning benchmarks (PIQA, ARC-Easy, ARC-Challenge, HellaSwag, Winogrande) and two language modeling tasks (WikiText2 (Merity et al., 2017) and C4 (Raffel et al., 2020)).

For PTQ baselines, we use a 256-sample RedPajama subset (seq length 2048) for AWQ, GPTQ, and SmoothQuant; OmniQuant follows its 40-epoch calibration, and SpinQuant is calibrated for 2 epochs. For QAT baselines, EfficientQAT adopts Block-AP (4096 RedPajama samples, 2 epochs) followed by E2E on Alpaca; BitDistiller uses a 4096-sample Alpaca subset for KD-based QAT; and

Table 2: Zero-shot evaluation of LLaMA-3.2 3B on five downstream tasks. We report accuracy (%) for PIQA, HellaSwag, Winogrande, ARC-c, and ARC-e, along with the average.

| LLaMA-3.2-3B | | PIQA | Hella. | Wino. | ARC-c | ARC-e | Avg |
|---|---|---|---|---|---|---|---|
| | bfloat16 | 77.47 | 73.62 | 69.61 | 45.90 | 71.71 | 67.67 |
| w2a16 | ParetoQ | 66.70 | 43.48 | 52.49 | 21.93 | 44.36 | 45.79 |
| | EfficientQAT | 70.02 | 57.07 | 59.35 | 34.13 | 58.92 | 55.89 |
| | BitDistiller | 70.65 | 57.42 | 59.78 | 34.71 | 58.34 | 56.18 |
| | **Bit-by-Bit (ours)** | 71.87 | 58.03 | 60.38 | 35.58 | 58.71 | **56.91** |
| w2a2 | ParetoQ | 51.80 | 25.76 | 48.78 | 23.55 | 27.53 | 35.48 |
| | EfficientQAT | 56.53 | 34.76 | 52.17 | 21.84 | 35.23 | 40.10 |
| | BitDistiller | 60.87 | 42.15 | 54.03 | 26.72 | 47.61 | 46.28 |
| | **Bit-by-Bit (ours)** | 66.00 | 49.30 | 56.91 | 31.40 | 54.00 | **51.52** |

ParetoQ is trained on 4096 RedPajama + 4096 Alpaca samples for 2 epochs, aligned to our budget (vs. 30B tokens in the original). Since these methods target weight-only quantization, we extend them with activation quantizers: online dynamic scaling for EfficientQAT, asymmetric clipping for BitDistiller, and 2-bit SEQ for ParetoQ. We train Bit-by-Bit on a 4096-sample subset of RedPajama. For weight-only quantization, the model precision is progressively reduced from `w8a16` to `w4a16` and then to `w2a16`, switching every two epochs, while splitting 10% of weight channels as detected by the metric. For weight–activation quantization, we first lower the weight precision to `w2a16`, then reduce the activation precision to `w2a2` progressively, splitting 10% of weight channels.

## 4.2 MAIN RESULTS

Table 1 reports perplexity results on WikiText2 and C4 under both weight-only (w2a16) and weight-activation (w2a2) settings. **Bit-by-Bit** consistently surpasses ParetoQ, EfficientQAT, and BitDistiller across model sizes and datasets. In w2a16, it requires fewer training tokens than ParetoQ, converges faster than BitDistiller, and achieves more stable training than EfficientQAT, e.g., reaching 11.25/17.41 PPL on WikiText2/C4 with LLaMA-3.2 3B. The advantage is even more pronounced in w2a2, where it reduces WikiText2 PPL on LLaMA-2 7B to 7.72, far below EfficientQAT (26.06) and BitDistiller (29.66). Zero-shot results (Table 2) further confirm its robustness: Bit-by-Bit achieves the best average accuracy under both w2a16 (56.91) and w2a2 (51.52), exceeding the strongest baseline by over 5 points in the latter. These results demonstrate Bit-by-Bit's effectiveness in preserving strong generalization under ultra-low precision.

## 4.3 ONCE-FOR-ANY-PRECISION EVALUATION

Our *once-for-any-precision* method produces models at multiple bit-widths. To validate the generality of this approach, we compare against MatQuant and OmniQuant on Mistral-7B. Specifically, we perform a single QAT run with Bit-by-Bit and directly apply the trained model to different bit-widths (w8a16, w4a16, w2a16). In contrast, the baseline OmniQuant requires separate training for each bit-width, while MatQuant also employs a one-shot QAT strategy for multi-bit adaptation. As shown in Table 3, our method achieves competitive or superior results under all settings. For w8a16 and w4a16, Bit-by-Bit matches the full-precision baseline with only marginal degradation, obtaining task averages of 73.51 and 73.21, respectively. More importantly, in the challenging w2a16 setting, Bit-by-Bit achieves a task average of 65.37 with C4 perplexity 10.73, substantially outperforming OmniQuant

Table 3: Evaluation of Mistral-7B under different quantization settings

| Mistral-7B | | | |
|---|---|---|---|
| Bits | Method | C4 ppl | Task avg |
| bfloat16 | | 8.24 | 73.99 |
| w8a16 | OmniQuant | 8.24 | 73.77 |
| | MatQuant | 8.43 | 73.46 |
| | **Bit-by-Bit (ours)** | 8.33 | 73.51 |
| w4a16 | OmniQuant | 8.47 | 73.62 |
| | MatQuant | 8.63 | 73.13 |
| | **Bit-by-Bit (ours)** | 8.79 | 72.21 |
| w2a16 | OmniQuant | 50.99 | 59.74 |
| | MatQuant | 13.05 | 65.99 |
| | **Bit-by-Bit (ours)** | 10.73 | 65.37 |

(59.74 / 50.99) and remaining on par with MatQuant (65.99 / 13.05). This demonstrates that a single QAT run suffices to deploy models at multiple bit-widths, eliminating the additional cost of retraining separate models for each configuration.

Table 4: Ablation study on Llama 3.2-1b on w2a16 setting, evaluation conducted on WikiText2 and 5 zero-shot tasks

| Block-wise | Progressive | Ocs | Metric | group size | WikiText2 ppl | Task avg | Memory |
|---|---|---|---|---|---|---|---|
| - | - | - | - | 32 | 1.7e3 | 35.09 | 0.33GB |
| ✓ | - | - | - | 32 | 31.88 | 40.87 | 0.33GB |
| ✓ | ✓ | - | - | 32 | 24.60 | 43.26 | 0.33GB |
| ✓ | ✓ | ✓ | Kurtosis | 32 | 22.43 | 43.69 | 0.36GB |
| ✓ | ✓ | ✓ | $w_{\max}$ | 32 | 20.37 | 44.26 | 0.36GB |
| ✓ | ✓ | ✓ | $x_{\max}$ | 32 | 19.07 | 44.30 | 0.36GB |
| ✓ | ✓ | ✓ | $|\mathbf{x}|_2 \cdot \overline{|w|}$ | 32 | 17.07 | 45.18 | 0.36GB |
| ✓ | ✓ | ✓ | $|\mathbf{x}|_2 \cdot \overline{|w|}$ | 64 | 30.26 | 40.66 | 0.34GB |
| ✓ | ✓ | ✓ | $|\mathbf{x}|_2 \cdot \overline{|w|}$ | 128 | 38.92 | 38.60 | 0.32GB |

## 4.4 ABLATION

We conduct a comprehensive ablation study of our proposed components on LLaMA3.2-1B, evaluating WikiText2 perplexity and the average score across five zero-shot tasks. As shown in Table 4, using block-wise loss yields substantially better results than end-to-end training with Negative Log-Likelihood. Training directly on w2a16 performs poorly, whereas adopting our progressive training strategy significantly improves convergence and accuracy. Incorporating outlier channel splitting (OCS) brings further gains. We evaluate several metrics for detecting outlier channels, including weight maximum ($w_{\max}$), activation maximum ($x_{\max}$), and kurtosis (DeCarlo, 1997; Nrusimha et al., 2024) which measures the "tailedness" of a distribution, and find that the combined weight–activation metric $|\mathbf{x}|_2 \cdot |\mathbf{w}|$ yields the best performance. While OCS slightly widens the weight matrix, the memory overhead remains modest (0.33GB $\rightarrow$ 0.36GB). We also examine the impact of group size: using group-128 saves only 0.04GB of memory but leads to a sharp degradation in performance that task accuracy falls from 45.18 to 38.60.

## 4.5 SPEED MEASUREMENT

We measure end-to-end decode speed (tokens/s) of Llama-3.2-1B under torch FP16 (w16a16), SpinQuant (w4a4), and our BIT-BY-BIT (w2a16/w2a2). For each sequence length (512–4k), we prefill the KV cache and report average decode speed over 256 tokens. Results show BIT-BY-BIT delivers the highest throughput across all lengths, with up to **1.95×** gain at short sequences and steady advantages at longer contexts.

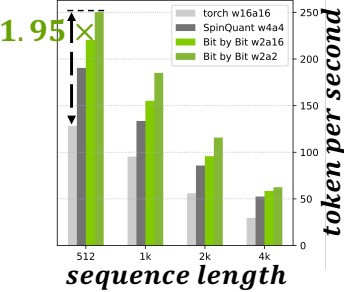

Figure 5: Decode throughput (tokens/s) at different sequence lengths.

## 5 CONCLUSION

We introduced BIT-BY-BIT, a stable low-bit QAT framework for LLMs that combines (i) progressive precision decay—reducing weight bits before activation bits in a block-wise schedule, (ii) a once-for-any-precision multi-target objective that trains a single model to operate at several bit-widths without retraining, and (iii) rounding-aware outlier-channel splitting that preserves the quantized output while shrinking rounding error. BIT-BY-BIT turns ultra-low-bit training into a coarse-to-fine adaptation problem, yielding robust convergence, practical deployment flexibility (one trained model, many precisions), and favorable accuracy–efficiency trade-offs.

# 6 ETHICS STATEMENT

We acknowledge and adhere to the ICLR Code of Ethics. We have carefully considered the ethical implications of our research and paper submission. Our work does not involve human subjects, and it does not make use of data sets that could raise privacy or security concerns. We have ensured that our methodology and applications do not introduce or perpetuate harmful biases, and we have taken care to document our data sources and experimental procedures to promote transparency and reproducibility. We have no known conflicts of interest or sponsorship to disclose.

# 7 REPRODUCIBILITY STATEMENT

We are committed to providing sufficient detail for the academic community to reproduce the results presented in this paper. All experiments were performed on a NVIDIA H800 GPU. We utilized the official implementations of all baseline methods where available, ensuring consistent environment configurations. Our evaluations were conducted on two major model families: the LLaMA series and the Mistral series. Performance was measured across seven standard benchmarks: Zero-Shot Reasoning: PIQA, ARC-Easy, ARC-Challenge, HellaSwag, and Winogrande; Language Modeling: WikiText2 and the C4 test set. We took measures to align the training cost across all QAT approaches for an unbiased evaluation. - EfficientQAT was first subjected to the Block-AP stage, utilizing a 4096-sample RedPajama subset over 2 epochs, and then proceeded to the E2E stage using the entire Alpaca dataset. - For BitDistiller, knowledge distillation was performed on a 4096-sample Alpaca subset synthesized by the teacher model. - ParetoQ's training budget was limited to 2 epochs, leveraging a combined dataset comprising a 4096-sample RedPajama subset and an equal-sized 4096-sample Alpaca subset. Furthermore, because these QAT baselines were inherently weight-only, we customized the activation quantization for each: EfficientQAT used a dynamic quantizer, BitDistiller relied on asymmetric clipping, and ParetoQ was equipped with a 2-bit SEQ quantizer. We used a 4096-sample subset of RedPajama in our Bit-by-Bit training process. In the process of Weight-Only Quantization, we incorporated the splitting of 10% of weight channels based on the metric at each step. In the process of Weight-Activation Quantization, we maintain the 10% channel splitting rule.

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

# APPENDIX

## A EXTENDED DISCUSSION

### A.1 THE USE OF LARGE LANGUAGE MODELS (LLMs)

A large language model was utilized for grammatical and stylistic refinement of the manuscript. Its role was strictly limited to text editing and polishing to enhance clarity. All research ideas, experimental design, and analytical content are the original work of the authors.

## A.2 BROADER IMPACTS

Our work advances ultra-low-bit quantization of large language models through a progressive training strategy with outlier channel splitting. By enabling stable training at 2-bit and below, **Bit-by-Bit** reduces the memory footprint and computational cost of LLMs by orders of magnitude. This improvement directly translates into lower inference latency, reduced energy consumption, and smaller carbon emissions, making the deployment of LLMs more sustainable.

Beyond efficiency, democratization is another key impact: with drastically reduced hardware requirements, powerful LLMs become accessible to a wider range of users and organizations, including those with limited computing resources. This may empower broader participation in AI research and applications, bridging the gap between well-funded institutions and smaller labs or industry players.

On the societal side, compressed LLMs can be deployed in edge scenarios such as mobile devices, offline environments, and privacy-sensitive settings, expanding the reach of AI to education, healthcare, and accessibility applications. However, lowering the barriers to deployment also amplifies risks of misuse, such as generating disinformation at scale or enabling harmful applications on inexpensive hardware. Mitigating these risks requires complementary safeguards, responsible governance, and continued community awareness.

Overall, we believe our work contributes to the ongoing effort of making LLMs greener, more efficient, and more inclusive, while highlighting the importance of balancing technological progress with responsible use.

## A.3 LIMITATIONS

While BIT-BY-BIT improves stability at ultra–low bits, it has several limitations. (i) We observe larger performance drops on the Qwen family, these models appear harder to quantize, leading to greater quantization error, and a deeper analysis is left for future work. (ii) The block-wise training schedule is less friendly to distributed training than end-to-end schemes, requiring nontrivial load-balancing and communication engineering. (iii) We have not extensively explored direct end-to-end progressive training; its convergence behavior and trade-offs remain open. (iv) We have not explored directions include learning layerwise schedules and split ratios automatically, extending to MoE and longer-context inference (e.g., KV-cache quantization), integrating hardware-aware mixed-precision search, and combining our training with lightweight distillation.

## B  EXTENDED AND DETAIL METHOD

### B.1  DIFFERENT PROGRESSIVE STRATEGIES

#### B.1.1  PRECISION PROGRESSIVE STRATEGIES

**(A) Weights → Activations (claimed in method).**   We first lower the *weight* precision to stabilize the network under weight noise, and only then reduce the *activation* precision:

$$(w8, a16) \rightarrow (w4, a16) \rightarrow (w2, a16) \rightarrow (w2, a8) \rightarrow (w2, a4) \rightarrow (w2, a2).$$

**(B) Alternating W/A.**   We interleave the bit reductions of weights and activations:

$$(w8, a16) \rightarrow (w8, a8) \rightarrow (w4, a8) \rightarrow (w4, a4) \rightarrow (w2, a4) \rightarrow (w2, a2).$$

**(C) Cyclic Precision (Kim et al., 2022)**   Unlike monotone schedules, cyclic precision alternates between $(k+1)$- and $k$-bit training before committing to $k$-bit. The idea is to leverage the smoother loss landscape of $(k+1)$-bit to recalibrate scales and reduce STE bias, while gradually adapting to the coarser $k$-bit lattice. A typical sequence is

$$(w16, a16) \rightarrow (w3, a16) \rightarrow (w2, a16) \rightarrow (w3, a16) \rightarrow (w2, a16) \cdots \rightarrow (w2, a2).$$

In practice, we first warm up from 8-bit down to $(k+2)$-bit, then run several short cycles between $(k+1)$ and $k$, and finally fine-tune at $k$-bit. This cyclic back-and-forth helps avoid representation collapse at ultra-low bits (e.g., 2-bit) by ensuring parameters remain quantizable on both lattices. While it introduces extra bit switches and hyperparameters, it often improves stability compared to a one-shot drop.

---

**Algorithm 2** Block-wise Progressive Strategy

---

1: **Input:** blocks $1..L$, stages $t = 1..T$, bits $\{b_t\}$, ratios $\{r_t\}$, bias $\alpha$
2: **for** $t = 1$ **to** $T$ **do**                                                              ▷ progressively lower precision
3:     Compute $p_j \propto (L+1-j)^\alpha$ and sample $\mathcal{S}_t$ with $|\mathcal{S}_t| = \lfloor r_t L \rfloor$
4:     **for** $j = 1$ **to** $L$ **do**
5:         **if** $j \in \mathcal{S}_t$ **then**
6:             Quantize block $j$ to bit $b_t$; *(others stay at previous bit)*
7:         **end if**
8:     **end for**
9:     (Optional) apply OCS to top-$r_\ell$ channels in selected blocks
10:    QAT for a fixed budget (steps/epochs) with short LR warmup
11: **end for**

---

**Empirical observations.**    We typically find Schedule (A) more stable (smoother loss/PPL decay, fewer divergence events), likely because it avoids simultaneous large shifts in both parameter and activation distributions. The alternating scheme can work but is more sensitive to optimizer and clipping hyperparameters and often requires longer warmup.

### B.1.2    BLOCK-WISE PROGRESSIVE STRATEGY

We adopt a stochastic, depth-aware curriculum over transformer blocks. Let the model have $L$ blocks indexed from input to output as $j = 1, \ldots, L$. At stage $t$ (with target bit $b_t$), we quantize only a subset $\mathcal{S}_t \subseteq \{1, \ldots, L\}$, sampled with a bias toward earlier blocks and with an increasing coverage over stages.

**Depth-biased sampling.**    Define a per-block sampling probability

$$p_j \ \propto \ (L + 1 - j)^\alpha, \qquad \alpha \geq 0,$$

so earlier blocks (small $j$) are more likely to be selected. Given a stage-wise coverage ratio $r_t \in (0, 1]$, we sample $|\mathcal{S}_t| = \lfloor r_t L \rfloor$ blocks without replacement according to $\{p_j\}$.

**Bit schedule.**    We follow a high-to-low bit curriculum, e.g.,

$$b_1 = 8 \ \to \ b_2 = 4 \ \to \ b_3 = 2,$$

and optionally apply the same scheme to activations after weights. The coverage ratio increases with $t$ (e.g., $r_t$ linear or cosine from $r_1 \approx 0.3$ to $r_T = 1.0$).

**Notes.**    (1) Depth bias ($\alpha$) and coverage growth ($r_t$) control stability/speed; we find $\alpha \in [0.5, 1]$ and linear $r_t$ robust. (2) This stochastic schedule avoids large simultaneous distribution shifts and is more kernel-friendly than fully per-step rebitting. (3) For a deterministic variant, select the first $\lfloor r_t L \rfloor$ blocks at each stage instead of sampling.

### B.2    MIXED-PRECISION OF DOWN-PROJECTION

As observed by (Chen et al., 2025), the inputs to the MLP down-projection (*FC2 Proj*) in Transformer blocks exhibit persistent activation outliers (high kurtosis). Under ultra–low-bit W/A quantization (e.g., W2A2), these heavy tails dominate the activation quantization error. To remove this bottleneck, we adopt a *layer-wise mixed-precision* scheme that raises the activation bit-width only for outlier-dominated sites while keeping the rest of the network at low precision. Concretely, we compute per-layer activation kurtosis $\kappa$ on a calibration set and mark layers with $\kappa > \tau$ as outlier-sensitive; for these layers we set $w2a4$ (with the same group-wise scaling as elsewhere), while all remaining layers use $w2a2$. This targeted relaxation substantially reduces activation quantization error—especially at coarse group sizes—while incurring minimal overhead and preserves the benefits of ultra–low-bit quantization in the rest of the model.

### B.3 LoRA FOR DISTRIBUTION-PRESERVING PROGRESSION

As illustrated in Fig. 4 (a), the higher-bit stage establishes a well-conditioned weight/activation distribution that serves as a strong initialization for subsequent lower-bit stages. To preserve this distribution while reducing precision progressively, we insert low-rank adapters (LoRA) (Hu et al., 2022) and restrict updates to these adapters rather than the full quantized backbone.

Concretely, when moving from bitwidth $b_t$ to $b_{t+1}$ ($b_{t+1} < b_t$), we freeze the backbone weights $W^{(t)}$ and optimize only a rank-$r$ perturbation

$$W^{(t+1)} = W^{(t)} + \alpha A^{(t)} B^{(t)^\top}, \qquad A^{(t)} \in \mathbb{R}^{d \times r}, \ B^{(t)} \in \mathbb{R}^{k \times r},$$

with the forward pass quantized as

$$W_q^{(t+1)} = Q_{s^{(t+1)}}(W^{(t)} + \alpha A^{(t)} B^{(t)^\top}).$$

To further stabilize the transition, we use a light distribution-matching regularizer that anchors first/second-order statistics of either weights or activations across stages, e.g.,

$$\mathcal{L}_{\text{dist}} = \left\| \mu(W_q^{(t+1)}) - \mu(W_q^{(t)}) \right\|_2 + \lambda \left\| \sigma(W_q^{(t+1)}) - \sigma(W_q^{(t)}) \right\|_2,$$

optionally combined with a KL term on layer activations. In practice we adopt small ranks ($r \in \{4, 8\}$) and reinitialize adapters at each stage. This *distribution-preserving* LoRA update significantly mitigates representation drift and reduces instability at ultra-low bits (e.g., 2-bit), while cutting trainable parameters to a $\frac{r(d+k)}{dk}$ fraction of full fine-tuning. After convergence, adapters are merged and requantized or discarded after re-estimating scales.

### B.4 SYMMETRIC MICROSCALING VIA SEQ

Our main pipeline uses asymmetric integers for simplicity, whereas microscaling formats (e.g., MXFP4/NVFP4) favor *symmetric* payloads with zero-point fixed at 0. To avoid the 2-bit degeneration to ternary under strict symmetric uniform grids, we adopt *Stretched Elastic Quantization (SEQ)* (Liu et al., 2025b), an LSQ-style amendment tailored for low-bit settings.

$$W_Q = \alpha \left( \frac{\left\lfloor \text{Clip}\left(\frac{W}{\alpha}, -1, 1\right) \cdot \frac{k}{2} - \frac{1}{2} \right\rfloor + \frac{1}{2}}{k} \right) \times 2,$$

which places centers at half-integers; for $b=2$ the normalized levels are $\left\{ -\frac{3}{4}, -\frac{1}{4}, \frac{1}{4}, \frac{3}{4} \right\}$. Here $\alpha \in$ FP8 is stored/rounded in FP8 per group, and $S_\text{T} \in$ FP32 is shared per tensor. The dequantized values are

$$\hat{W} = S_\text{T} \cdot W_Q = S_\text{T}\, \alpha \cdot \left(n + \frac{1}{2}\right), \quad n \in \left\{ -\frac{k}{2}, \ldots, \frac{k}{2} - 1 \right\}.$$

At $b=2$, the LUT becomes

$$\mathcal{C}_{\text{SEQ-2b}} = S_\text{T}\, \alpha \cdot \{-1.5, -0.5, 0.5, 1.5\}.$$

This keeps a zero-point–free symmetric path, matches NVFP4's FP8 group scale + FP32 master scale, and fully uses all four codes at 2-bit.

### B.5 MUON FOR LOW-BIT QAT: TRAINING DYNAMICS

We investigated whether the *Muon* (Liu et al., 2025a; Park et al., 2025) optimizer can stabilize training dynamics in ultra–low-bit QAT. In our pipeline, the per-group scale and zero-point are computed *online*; thus the only trainable variables are the full-precision 2D weight matrices, while quantizer statistics are not explicitly optimized.

**Setup.** We keep the learning-rate schedule, batch size, and clipping identical to the AdamW baseline, and apply STE for quantization with progressive bit reduction.

**Observation.** Across models and bit settings, Muon did not yield consistent gains over AdamW: convergence speed and final perplexity were comparable or slightly worse, and we observed larger short-horizon oscillations near quantization thresholds in some layers.

**Possible causes (hypotheses).** (i) Online rescaling induces non-stationary curvature that weakens Muon's preconditioning benefits under STE noise; (ii) gradient signals are dominated by rounding discontinuities at ultra–low bits, reducing the utility of curvature-aware updates; (iii) block/group-wise statistic updates interact with momentum, amplifying drift.

**Next steps.** We will explore (a) using Muon only on LoRA adapters while freezing the backbone; (b) scale-aware trust-region or gradient clipping around threshold crossings; (c) layer-wise Muon/AdamW hybrids. At present, Muon does not provide a clear advantage for our low-bit QAT setting.

