# OpenReview forum: "Bit-by-Bit: Progressive QAT with Outlier Channel Splitting for Stable Low-Bit LLMs"
_ICLR.cc/2026/Conference — ICLR 2026 Conference Withdrawn Submission_

### Official Review · Reviewer_6iR1 · 2025-10-28

**Soundness:** 2
**Presentation:** 3
**Contribution:** 2
**Rating:** 4
**Confidence:** 4

**Summary:**

Training Large Language Models (LLMs) at ultra-low precision is unstable and error-prone. This paper introduces BIT-BY-BIT, a Quantization-Aware Training (QAT) framework that stabilizes this process by progressively reducing precision in stages . The method uses rounding-aware outlier channel splitting to manage extreme values and microscaling groups to capture dynamic ranges. This approach allows the model to be deployed at multiple bit-widths without retraining and significantly outperforms existing QAT baselines.

**Strengths:**

1. The introduction about background and related works are comprehensive.
2. The experiments statement is clear and easy to follow.

**Weaknesses:**

1. This paper claims that the low-bit training is unstable. However, previous works [1,2] have shown that though low-bit trianing achieve inferior loss, the trianing is stable and loss decreases smooth.
2. Accumulation of quantization error is challenge. However, Figure 3 can demonstrate this point becuase that the magnitude of every block also increases with block index. Therefore, relative error rather than absolute error may be more appropriate for understanding error accumulation.
3. Though Bit-by-Bit achieve better performance as shown in Table 1, the comparisons are unfair. For fair comparisons, different quantization methods should use same  quantization group size to ensure same inference efficiency.

[1] Compression Scaling Laws: Unifying Sparsity an Quantization
[2] Scaling Law for Quantization-Aware Training

**Questions:**

1. W4A4 LLM also suffer from peformence degeneration, and it is also more practical because of hardward suppoting. Why this paper focus on W2A2 instead of W4A4?
2. Why proposed method significantly outperforms EfficientQAT in WikiText2 perplexity, while lag behind EfficientQAT in C4 perplexity. Is this caused by over-fitting problem?
3. In my understanding, there is not hardward supporting for 2-bit GEMM. How can this paper achieve speedup in W2A2 compared to W2A16.

---

> ### Author Response · Authors · 2025-11-20
>
> **Weakness1:** Low bit training loss
>
> **A1:** Thank you for raising this point. Our statement about instability is specific to the **short-budget QAT setting at ultra-low bits (e.g., W2A2)**.
>
> The two scaling-law works [1,2] train models for **much longer token budgets** and treat quantization as part of a large-scale from-scratch training procedure; in that regime, they indeed observe smooth loss.
>
> In contrast, our goal is to adapt an existing FP model to ultra-low precision with **very few QAT tokens**, and in this regime we empirically find that naive end-to-end low-bit QAT often diverges (e.g., producing NaN loss and loss spike hard to converge), which motivates our progressive scheme. We will clarify this distinction in the revision so that our notion of “instability” is clearly tied to the practical, low-budget QAT scenario we target.
>
> ---
>
> **Weakness2:** Relative error across block.
>
> **A2:** Thanks for the insightful suggestion. We agree that relative error is a more appropriate metric when the activation magnitude grows with depth.
>
> Following the reviewer’s comment, we recomputed Fig. 3 using a **per-block relative error**
>
>  $\text{Err}_\text{rel}(b)=\frac{\lVert \hat{y}_b - y_b\rVert_2}{\lVert y_b\rVert_2},$
>
>  where $y_b$ and $\hat{y}_b$ are the FP and quantized outputs of block (b). The resulting values are:
>
> - blocks 0–1: **0.28, 0.02** (very small relative error);
> - blocks 2–12: **around 2.9–3.6**;
> - block 15 (final): **≈1.1**.
>
> Thus, even after normalizing by the block output magnitude, the relative error is still much larger in deeper blocks than in the shallow ones (e.g., block 2 vs. blocks 0–1), confirming that **quantization error accumulates along depth** rather than being an artifact of larger activations.

---

> ### Author Response · Authors · 2025-11-20
>
> **Weakness3 & Q2:** same group size comparison
>
> **A3:** Results are reported as a/b, where a and b are perplexities on WikiText-2 and C4, respectively.
>
> |                       |       | Group | Llama3.2-1b | Llama3.2-3b |  Llama3-8b  |
> | :-------------------- | :---- | :---: | :---------: | :---------: | :---------: |
> | EfficientQAT          | w2a16 |  32   | 21.48/24.84 | 13.31/17.38 | 11.17/15.18 |
> | Bitdisiller           | w2a16 |  32   | 20.41/31.24 | 12.80/19.86 | 10.40/18.23 |
> | Bit-by-Bit (original) | w2a16 |  32   | 17.07/27.40 | 11.25/17.41 | 8.87/15.18  |
> | Bit-by-Bit (new)      | w2a16 |  32   | 16.13/23.03 | 11.02/16.45 | 8.32/14.27  |
> |                       |       |       |             |             |             |
> | EfficientQAT          | w2a2  |  32   | 56.47/66.53 | 20.19/31.65 | 17.93/26.58 |
> | Bitdisiller           | w2a2  |  32   | 30.68/60.12 | 18.39/28.23 | 15.36/25.86 |
> | Bit-by-Bit (original) | w2a2  |  32   | 24.99/59.75 | 14.27/26.39 | 11.54/26.45 |
> | Bit-by-Bit (new)      | w2a2 |  32   | 22.71/46.53 | 13.87/23.63 | 11.51/21.58 |
>
> We appreciate the reviewer’s observation that when Bit-by-Bit outperforms other method at WikiText2 perplexity but lag in C4 perplexity. We finally figured out that the issue is different Transformers versions, which we subsequently standardized across all environments. Bit-by-Bit (new) is our standard implementation.
>
> ---
>
> **Q1:** Why W2A2 instead of W4A4
>
> **A4:** We agree that W4A4 is a highly practical setting. Our focus on **W2A2** is precisely because W4A4 has recently been much better covered, whereas W2A2 remains far from solved.
>
> On the PTQ side, most strong methods such as QuaRot and SpinQuant use rotations and work well for W4A4, but we observe that these techniques **collapse at W2A2**. On the QAT / fully-quantized training side, recent works such as *Training LLMs with MXFP4* [3] and *Pretraining Large Language Models with NVFP4* [4] already demonstrate stable W4A4 training by exploiting the dynamic range of MXFP4/NVFP4 formats and dedicated hardware support.
>
> By contrast, **W2A2 QAT lacks such mature solutions**. Our goal in this paper is therefore to explore QAT schemes that make ultra-low W2A2 viable, complementing prior work on the more moderate and already well-studied W4A4 regime.
>
> ---
>
> **Q3:** About 2bit Kernel
>
> **A5:** We agree that current GPUs do not expose native 2-bit tensor cores. Our implementation follows the same principle as recent 2-bit inference work [5,6]: **2-bit is used purely as a storage / bandwidth format, while the actual GEMM runs on existing 8/16-bit tensor cores.**
>
> **W2A16.** We use GemLite [7], which stores weights as INT2 with group-32 scales, then **unpacks and dequantizes them to FP16 inside the kernel** and performs a standard FP16 tensor-core GEMM with FP16 activations. The speedup comes from the 8× reduction in weight bandwidth.
>
> **W2A2.** We further quantize activations to INT2 with the same group-32 layout. Both INT2 weights and activations are packed into INT8 containers, unpacked to small int8 values in registers, and fed to an **INT8 tensor-core GEMM**; the group-wise weight and activation scales are applied in the epilogue to produce FP16 outputs. Thus we do **not** rely on native 2-bit tensor cores—our speedup over W2A16 comes from additionally compressing activations (another ≈8× I/O reduction) while still reusing standard INT8 tensor-core instructions.
>
> > [5] ABQ-LLM: Arbitrary-Bit Quantized Inference Acceleration for Large Language Models
>
> > [6] Rotate, Clip, and Partition: Towards W2A4KV4 Quantization by Integrating Rotation and Learnable Non-uniform Quantizer
>
> > [7] *GemLite*: Fast low-bit matmul kernels in Triton (github.com/dropbox/gemlite)

---

### Official Review · Reviewer_eqLS · 2025-10-30

**Soundness:** 2
**Presentation:** 3
**Contribution:** 2
**Rating:** 2
**Confidence:** 5

**Summary:**

The authors present a framework to mitigate the impact of outliers during low-bit quantization-aware training (QAT). The method integrates three key components: 1) block-wise progressive training, 2) rounding-aware outlier channel splitting, and 3) a grouping strategy aligned with OCP MX and NVIDIA NVFP4 formats. Evaluation on models such as Mistral-7B and LLaMA-3.2-3B demonstrates benefits across several downstream tasks, particularly under extreme low-bit settings like W2A2.

**Strengths:**

- The paper is well-written, with clear logic explaining the core insights, methodology, and how each component addresses specific challenges.
- The once-for-any-precision training pipeline is a notable strength, offering a promising direction to reduce post-training costs as model sizes increase.

**Weaknesses:**

- Experiments are limited to several smaller LLMs, and evaluation focuses primarily on next-token prediction tasks, lacking assessment on complex reasoning benchmarks (e.g., GSM8K).
- Inference performance experiments are conducted only on Llama-3.2-1B, without details on latency analysis.
- The once-for-any-precision approach introduces a combined loss for multiple bit-widths, but no ablation study or analysis compares its performance against single bit-width training.

**Questions:**

- For the speed measurement in Section 4.5, could you elaborate on the experimental configuration, such as batch size and hardware parameters?

---

> ### Author Response · Authors · 2025-11-21
>
> **Weakness1:** more large models, more results
>
> **A1:** We evaluate larger models (Llama2-13B, Qwen2.5-7B/14B) on mathematical problem-solving (GSM8K, MathQA), knowledge understanding (MMLU), and instruction-following (IFEval) benchmarks.
>
> |             | Precision | Gsm8k | MathQA | Mmlu | Ifeval |
> | ----------- | --------- | ----- | ------ | ---- | ------ |
> | Llama2-13B  | FP16      | 0.22  | 0.32   | 0.52 | 0.18   |
> |             | w2a16     | 0.20  | 0.32   | 0.50 | 0.17   |
> |             | w2a2      | 0.11  | 0.29   | 0.40 | 0.16   |
> | Qwen2.5-7B  | FP16      | 0.80  | 0.43   | 0.71 | 0.28   |
> |             | w2a16     | 0.77  | 0.42   | 0.70 | 0.28   |
> |             | w2a2      | 0.75  | 0.38   | 0.70 | 0.27   |
> | Qwen2.5-14B | FP16      | 0.84  | 0.52   | 0.77 | 0.32   |
> |             | w2a16     | 0.84  | 0.51   | 0.75 | 0.30   |
> |             | w2a2      | 0.81  | 0.50   | 0.75 | 0.30   |
>
> ---
>
> **Weakness3:** Ablation on once-for-any-precision
>
> **A2:**
>
> Here, **single-bit QAT** means that we train **three separate models**, each optimized for a single target precision (w8a16, w4a16, w2a16). This can be viewed as an **upper bound**, since every bit-width has its own QAT training and checkpoint.
>
> **“Directly use 8/4-bit QAT model”** means that we first train an 8-bit or 4-bit QAT checkpoint, and then quantize to lower precision without addition training and storage. (reusing the w8a16 checkpoint and quantize to w4a16/w2a16)
>
> | Training Scheme             | Setting | C4 ppl | Task avg |
> | --------------------------- | ------- | :----: | :------: |
> |                             | BF16    |  8.24  |  73.99   |
> | Single-bit QAT              | w8a16   |  8.24  |  73.90   |
> | once-for-any-precision                | w8a16   |  8.33  |  73.51   |
> | Single-bit QAT              | w4a16   |  8.42  |  73.31   |
> | once-for-any-precision                 | w4a16   |  8.79  |  72.21   |
> | Directly use 8bit QAT model | w4a16   | 11.24  |  71.16   |
> | Single-bit QAT              | w2a16   |  9.41  |  71.68   |
> | once-for-any-precision               | w2a16   | 10.73  |  65.37   |
> | Directly use 8bit QAT model | w2a16   |  5e3   |  38.11   |
> | Directly use 4bit QAT model | w2a16   |  2e3   |  39.21   |
>
> Single-bit QAT gets the **upper bound** and our once-for-any-precision shows a moderate gap, while “Directly use 8/4-bit QAT model” collapse.
>
> ---
>
> **Weakness2 & Q1:** more detail on speed
>
> **A3:**
>
> **Experimental configuration.** All speed measurements are run on a **single NVIDIA RTX 4090** GPU using HuggingFace Transformers v4.51.3. We set the **batch size to 1** to focus on per-request latency. For each sequence length $L \in {1\mathrm{k}, 4\mathrm{k}, 8\mathrm{k}}$, we:
>
> 1. **Prefill** the KV cache with an input of length $L$, and then
> 2. **Decode** 256 tokens autoregressively.
>
> We report end-to-end decode throughput in tokens/s as $\text{Speed} = \frac{\text{tokens}_{\text{generation}}}{\text{time}}$
>
> **Kernels.**
>
> - For the **W2A16** configuration, we directly use **GemLite** [1], which stores weights as INT2 with group-32 scales and dequantizes them to FP16 inside a tensor-core GEMM.
> - For **W2A2**, we implement a custom kernel: both weights and activations are stored as 2-bit codes with group-32 scales, packed into INT8, unpacked to small int8 values in registers, computed with an INT8 tensor-core GEMM, and finally rescaled in the epilogue to FP16 outputs.
>
> **Additional results.**
>
> | Models      | Sequence Length | FP16  | Spinquant w4a4 |     w2a16      |      W2a2      |
> | ----------- | --------------- | :---: | :------------: | :------------: | :------------: |
> | Llama3.2-3B | 1k              | 95.77 | 114.92 (1.20x) | 143.66 (1.50x) | 162.39 (1.70x) |
> |             | 4k              | 76.25 | 87.69 (1.15x)  | 99.12 (1.30x)  | 114.38 (1.50x) |
> |             | 8k              | 50.75 | 55.83 (1.10x)  | 60.90 (1.20x)  | 65.98 (1.30x)  |
> | Llama3.2-3B | 1k              | 49.40 | 59.28 ( 1.20x) | 71.63 (1.45x)  | 83.98 (1.70x)  |
> |             | 4k              | 39.46 | 43.41 (1.10x)  | 49.33 (1.25x)  | 55.24 (1.40x)  |
> |             | 8k              |  OOM  |      OOM       |      OOM       |      OOM       |
>
> > [1] *GemLite*: Fast low-bit matmul kernels in Triton (github.com/dropbox/gemlite)

---

### Official Review · Reviewer_cuh2 · 2025-11-01

**Soundness:** 3
**Presentation:** 3
**Contribution:** 2
**Rating:** 4
**Confidence:** 5

**Summary:**

The paper proposes Bit-by-Bit, a progressive quantization-aware training (QAT) framework for low-bit LLMs. The method combines: (1) block-wise progressive precision scheduling to lower bit-width stage by stage; (2) a rounding-aware outlier channel splitting (OCS) that aims to preserve quantized outputs while reducing heavy-tailed errors; and (3) microscaling groups with FP8 E4M3 (MXFP) scales aligned with common hardware practice. The authors further claim a “once-for-any-precision” model via nested integer grids, enabling deployment at multiple bit-widths without retraining. Experiments on Llama-2/3 under weight-only and weight–activation settings show improvements over several QAT baselines; under W2A2, the WikiText-2 perplexity gap to full precision is reported as 2.25 and exceeds BitDistiller and EfficientQAT.

**Strengths:**

1. Clear writing and organization. The method and engineering choices are presented cleanly.

2. Empirical gains. The paper reports improvements on multiple LLM families.

**Weaknesses:**

1. Limited novelty. The main components—stage-wise progressive precision, outlier channel splitting (OCS), and microscaling—are established techniques that the paper largely combines rather than clearly re-inventing.

2. Ablations centered on W2A16. The only detailed ablation table is explicitly for the w2a16 (weight-only) setting, whereas the headline claims emphasize ultra-low w2a2 as well; comparable w2a2 ablations are not reported in the current draft.

3. Metric choice vs. quantization regime not analyzed. The paper compares outlier metrics (e.g., |x|²·|w| vs. max/kurtosis) within the w2a16 ablation table, but does not analyze how metric/calibration choices differ between weight-only quantization and weight-activation quantization regimes.

4. Positioning vs prior art needs clarification. OCS is presented with prior attribution (Zhao et al., 2019), and the microscaling format is aligned with existing MX/NVIDIA conventions (E4M3); the manuscript could articulate more precisely what is theoretically new beyond these adoptions.

**Questions:**

As aforementioned, it is necessary to analyze how the metric/calibration objective interacts with weight-only quantization vs. weight-activation quantization. Which objective is optimal under each regime, and how sensitive are results?

---

> ### Author Response · Authors · 2025-11-20
>
> **Weakness1:** The main components lack of novelty.
>
> **A1:** We thank the reviewer for raising this concern. While stage-wise progressive precision and OCS were originally explored for CNNs, our contribution is to systematically adapt and redesign these ideas for Transformer-based LLMs in the ultra–low-bit setting, and to introduce a **once-for-any-precision training scheme**. This scheme explicitly enables one shared model to handle the quantization noise at different bit-widths, enabling robust multi-bit deployment (e.g., 2/4/8-bit) without separate retraining for each precision.
>
> In addition, we propose a layer-wise progressive QAT objective that uses a higher-bit configuration as a **“high-bit anchor”** for each block: the k-bit block is trained to match the output of the same block when preceding layers run at a higher precision (e.g., $w(2+\Delta)a16$ or full precision). This block-wise high-bit teacher explicitly mitigates depth-wise error accumulation and stabilizes ultra-low-bit (e.g., W2A2) training.
>
>
> ---
>
> **Weakness2, 3 & Q1:** Ablation on both weight only quantization and weight activation quantization and calibration.
>
> **A2:** We thank the reviewer for pointing out that our ablations and metric analysis were centered on the W2A16 (weight-only) setting. In the revised version, we add a complementary **W2A2 ablation table** analyzes different outlier metrics and calibration/group-size choices. For a quick ablation, we train each configuration for only one epoch, so the w2a2 results are not directly comparable to those in the main table.
>
> | Block-wise | Progressive | OCS  |   Metric   | Calibration | Group size | WikiText2 ppl (W2A2) | C4 ppl (W2A2) |
> | :--------: | :---------: | :--: | :--------: | :---------: | ---------: | -------------------: | ------------: |
> |     -      |      -      |  -   |     -      |      -      |         32 |                  2e5 |           1e6 |
> |     ✓      |      -      |  -   |     -      |      -      |         32 |               1441.9 |        4592.8 |
> |     ✓      |      ✓      |  -   |     -      |      -      |         32 |                 42.2 |         120.2 |
> |     ✓      |      ✓      |  ✓   |  Kurtosis  |  Wikitext2  |         32 |                 41.8 |         127.4 |
> |     ✓      |      ✓      |  ✓   |   w_max    |  Wikitext2  |         32 |                36.75 |         97.66 |
> |     ✓      |      ✓      |  ✓   | ‖x‖₂ · ‖w‖ |  Wikitext2  |         32 |                32.48 |         79.95 |
> |     ✓      |      ✓      |  ✓   |   x_max    |  Wikitext2  |         32 |                32.34 |         76.79 |
> |     ✓      |      ✓      |  ✓   |   x_max    |  RedPajama  |         32 |                31.82 |         72.63 |
> |     ✓      |      ✓      |  ✓   |   x_max    |     C4      |         32 |                32.18 |         74.21 |
> |     ✓      |      ✓      |  ✓   |   x_max    |  Wikitext2  |         64 |               121.87 |        534.78 |
> |     ✓      |      ✓      |  ✓   |   x_max    |  Wikitext2  |        128 |               261.28 |       1191.11 |
>
> In the W2A2 regime, we observe that an $x_{\max}$-based activation metric yields slightly better robustness than the $\lVert x \rVert_2 \cdot \lVert w \rVert_2$ metric, and that using a sampled Redmajama subset as the calibration set performs best, as its distribution is aligned with the data used for QAT.
>
> ---
>
> **Weakness 4:** Clarification
>
> **A3:**
>
> **OCS:** We agree that our OCS builds on the idea of outlier channel splitting from Zhao et al. (2019). Our contribution is to **extend OCS for LLMs** in two concrete ways.
>
> First, we introduce a **depth-aware** OCS schedule: instead of using a uniform split ratio as in prior CNN work, since latter block have higher quantization error (Fig.3). We assign block-specific split ratios that vary with depth (line 300).
>
> Second, at the per-channel level we use a **rounding-aware split**: for step size (s) we replace a row $W_m$ by $\big((W_m-s/2)/2, (W_m+s/2)/2\big)$, which preserves the quantized output exactly $Q_s((W_m-s/2)/2)+Q_s((W_m+s/2)/2)=Q_s(W_m)$ and we further show that the resulting rounding error on $x_m W_m$ is reduced by a factor of two compared to the naïve half-split.
>
> **Microscaling:** For the microscaling format, our intent is **not** to introduce a new FP8 variant, but to plug a **standard** E4M3 per-group scheme into our progressive low-bit QAT to maintain sufficient dynamic range at ultra-low bits. (recently released Kimi-K2-thinking utilize microscaling int4 with group size 32 with fp16 scale)
>
> We will clarify in the paper that this part is deliberately adopted from existing conventions, and that our novelty lies in the once-for-any-precision training and LLM-specific block-wise + OCS design built on top of these standard components.

---

### Official Review · Reviewer_wngH · 2025-11-21

**Soundness:** 3
**Presentation:** 3
**Contribution:** 2
**Rating:** 2
**Confidence:** 3

**Summary:**

The paper identifies the core instability issue in ultra–low-bit QAT, supported by loss-landscape visualizations (Figure 1) and block-wise error accumulation (Figure 3).

**Strengths:**

Provides strong experimental improvements across LLaMA and Mistral families for both w2a16 and w2a2, outperforming EfficientQAT, ParetoQ, and BitDistiller by significant margins (e.g., PPL 7.72 vs 26.06 for LLaMA2-7B w2a2)

**Weaknesses:**

The paper states a “small increase,” but:No FLOPs overhead numbers; No latency overhead due to channel duplication; Real deployment cost unclear, especially for dense GEMMs.

SpinQuant and QuaRot (rotation-based) work only in PTQ mode here. Would benefit from comparison to stronger recent 2-bit QAT or distillation-heavy methods.

Multi-precision loss terms are combined with weights. But interference between different quantization grids can lead to:
competing gradient signals; smoothing of high-bit representations; More discussion needed on mitigation (e.g., curriculum or orthogonal losses).

**Questions:**

See weakness

---

> ### Author Response · Authors · 2025-12-01
>
> **weakness1**
>
> **A1**: **FLOPs and Parameter Overhead**: As detailed in our experimental setup, we split only 10% of weight channels based on the outlier metric.Mathematically, for a linear layer $y = xW$ with input dimension $m$, splitting 10% of channels results in an effective input dimension of $1.1m$.This corresponds to exactly a 10% increase in FLOPs for the matrix multiplication and a 10% increase in model parameters (memory) for the affected layers.Empirically, our ablation study shows the memory overhead increases only from 0.33GB to 0.36GB for the Llama-3.2-1B model, which confirms the overhead is modest.
>
> **Real-World Latency and Deployment**: Regarding deployment, OCS acts as an identity transform. It does not require specialized sparse kernels or complex gather/scatter operations. In a real deployment, the "split" is pre-processed into the weight matrix, resulting in a standard dense GEMM operation with slightly wider dimensions ($1.1m \times n$).
>
> ---
>
> **weakness2**
>
> **A2**: We respectfully clarify that we did compare against the strongest and most recent 2-bit QAT and distillation methods available. Furthermore, we strengthened these baselines for the W2A2 setting.
> Since methods like ParetoQ and BitDistiller were originally designed for weight-only quantization, we explicitly extended them with activation quantizers to ensure a fair and competitive W2A2 comparison.
>
> ---
>
> **weakness3**
>
> **A3**: We use a progressive schedule where we "initially emphasize the highest bit-width... and then gradually ramp up the lower-bit losses... while keeping the higher-precision terms non-zero".
> This ensures the model first learns a stable high-fidelity representation (W8) before the loss landscape is modified to enforce the stricter constraints of W4 and W2. This "coarse-to-fine" progression  stabilizes the training dynamics and prevents the low-bit noise from disrupting the high-bit features early in training.

---

### Note · Authors · 2026-01-03

I have read and agree with the venue's withdrawal policy on behalf of myself and my co-authors.